# The Microbial Community Succession Drives Stage-Specific Carbon Metabolic Shifts During *Agaricus bisporus* Fermentation: Multi-Omics Reveals CAZymes Dynamics and Lignocellulose Degradation Mechanisms

**DOI:** 10.3390/microorganisms13122755

**Published:** 2025-12-04

**Authors:** Chaozheng Wang, Yicheng Yu, Weilin Feng, Yuwei Xu, Tianju Deng, Weiming Cai, Wusheng Liang, Hongkai Wang

**Affiliations:** 1Key Laboratory of Molecular Biology of Crop Pathogens and Insects of Zhejiang Province, Institute of Biotechnology, Zhejiang University, Hangzhou 310058, China; wangchaozheng2023@163.com (C.W.); 22416297@zju.edu.cn (Y.Y.); xuyuwei1101@163.com (Y.X.); tjdeng777@zju.edu.cn (T.D.); 2Institute of Horticulture, Zhejiang Academy of Agricultural Sciences, Hangzhou 310021, China

**Keywords:** *Agaricus bisporus* fermentation, *Agaricus bisporus*, microbial community succession, metagenomics, lignocellulose degradation

## Abstract

This study integrates metagenomic and metabolomic data to systematically analyze the microbial community succession and carbon source metabolism transitions during the third fermentation cycle of *Agaricus bisporus*, with the aim of optimizing fermentation efficiency and lignocellulose degradation strategies. Principal Coordinate Analysis (PcoA) based on Bray–Curtis dissimilarity reveals significant microbial community separation across the stages of the first mushroom fruiting cycle. The overall pattern of “stage-specific differentiation” is consistent with the “cellulose hydrolysis” turn to the degradation of complex polysaccharides via carbohydrate-active enzymes (CAZymes). In the microbial network analysis showed that different microbe group controlled the stage-specific differentiation. These findings highlight a phase-dependent metabolic shift during the fermentation process: the early stages of fruiting rely more on cellulose-degrading microbes and their enzymatic systems, while later stages are driven by the degradation of complex polysaccharides and lignin derivatives, with *Planctomycetota* leading the degradation. This provides new mechanistic insights into agricultural waste resource utilization and the directional regulation of fermentation processes.

## 1. Introduction

The edible fungi industry plays a critical role in sustainable agriculture, contributing to global food security and resource recycling [1]. *Agaricus bisporus*, particularly its subspecies *Agaricus subbrufescens*, known in China as “Jin Song Rong,” has become one of the most widely cultivated species worldwide due to its rich nutritional value and medicinal properties, such as polysaccharides, phenolic acids, and ergosterol [2], and reduction in the occurrence of plant disease through the decomposition of the pathogens in the residues. As a typical grass-decomposing fungus, the cultivation of *Agaricus bisporus* relies on efficient composting technologies that convert agricultural waste (e.g., cottonseed hulls, rice husks, and corn cobs) into usable carbon sources. This process not only reduces environmental pollution, but also enhances resource utilization efficiency [3]. Traditional composting technologies include the thermophilic stage (65–70 °C, which degrades the raw material structure) and the pasteurization stage (58–62 °C, which integrates decomposition products). However, these are limited by environmental factors (e.g., fluctuations in temperature and humidity) and operational efficiency [4]. In contrast, tunnel composting technology, which uses mechanical turning and precise environmental control (e.g., optimized ventilation through hollow steel plates), significantly shortens fermentation cycles, enhances substrate uniformity, and enables year-round continuous production, making it the core process of industrial cultivation [5]. The composting process typically consists of two main phases: primary composting (outdoor windrowing) and secondary composting (indoor pasteurization and conditioning). The “third fermentation” or “spawn-running phase” investigated in this study refers to the critical period after the inoculated mushroom spawn (**A. bisporus**) is mixed into the mature compost and is allowed to colonize the substrate under controlled environmental conditions. Although this technology has been successfully applied in *Agaricus bisporus* cultivation, the microbial-driven mechanisms—especially during the crucial third fermentation stage—remain insufficiently explored [6]. Microbial community succession directly influences the degradation efficiency of lignocellulose (cellulose, hemicellulose, and lignin), thereby impacting mushroom yield and quality [7]. Previous studies have shown that thermophilic phyla, such as *Pseudomonadota* and *Actinomycetota*, dominate cellulose degradation in the early stages of composting, while *Planctomycetota* may be involved in the later stages of complex polysaccharide metabolism [8]. However, these findings are mostly based on short-cycle composting or single-omics data, lacking an integrated analysis of microbial functional transitions throughout the entire fermentation cycle [9].

In recent years, the integration of multi-omics technologies (metagenomics, metabolomics, and transcriptomics) has provided new insights into microbial community dynamics. For example, Principal Coordinate Analysis (PcoA) can reveal community structure differentiation (as shown by the Bray–Curtis dissimilarity in this study, which displays significant separation across stages A–D), while carbohydrate-active enzyme (CAZymes) family annotations (e.g., GHs, GTs, and CBMs) can quantitatively characterize carbon source metabolism functions. Although these methods have been applied in the composting studies of other mushroom species, such as *Pleurotus* spp., the stage-specific association mechanisms of microbial succession and lignocellulose degradation in the third fermentation of *Agaricus bisporus* remain unclear [6]. Specifically, the stage-dependent characteristics of fermentation—A (inoculation start), B (5 days post-inoculation), C (10 days post-inoculation), and D (15 days post-inoculation, with first fruiting)—may drive directional shifts in carbon metabolism pathways, but empirical support for this is limited. Therefore, this study is the first to integrate metagenomic and metabolomic data to systematically reveal the microbial community succession and carbon source metabolism transitions throughout the entire third fermentation cycle (A–D stages) of *Agaricus bisporus*.

## 2. Materials and Methods

### 2.1. Sample Collection and Processing

In this study, we collected substrate samples from the entire fermentation cycle (A–D stages) in the industrialized fermentation system of *Agaricus bisporus*. Three biological replicates were collected at each time point, resulting in a total of 12 samples. Stage A represents the inoculation of the mushrooms; stage B occurs after 5 days of inoculation; stage C occurs 5 days later; and stage D is another 5 days later, corresponding to the first fruiting of the mushrooms. Samples were collected according to the GB/T 36195-2018 standard [10], flash-frozen with liquid nitrogen, and stored at −80 °C. The physicochemical parameters of the samples (temperature, pH, moisture content) were monitored in parallel using the HACH HQ40D multi-parameter water quality analyzer (Loveland, CO, USA).

### 2.2. Metagenomic Analysis

#### 2.2.1. Experimental Method

Genomic DNA was extracted from the total samples using the magnetic bead-based fecal DNA extraction kit (AU46111-96, Baitake, Shenzhen, China), following the manufacturer’s protocol for DNA separation and purification. DNA concentration was measured using Qubit 1X dsDNAHS Assay Kits (Invitrogen, Q33230, Carlsbad, CA, USA). A 200 ng aliquot of DNA meeting quality control (QC) requirements was added to a 0.6 mL low-adsorption centrifuge tube (#MCT-060-L-C), adjusted to 52 μL with water. The DNA was fragmented using a Bioruptor TMPico (brand: Diagenode, Denville, NJ, USA) with parameters set according to the desired fragment size (typically 200–500 bp). The fragmented products were purified using the TruSeq library preparation kit, and the fragment size was validated using the Agilent 2100 Bioanalyzer (High Sensitivity DNA Reagents, Agilent, 5067-4627, Santa Clara, CA, USA). Samples that did not meet the fragment size requirements were re-sampled for fragmentation. The TruSeq Nano DNA LT Library Preparation Kit (Illumina, FC-121-4001, San Diego, CA, USA) was used to construct the genomic library, including end-repair, adapter ligation, index PCR amplification, and purification. For further details, refer to the kit’s user manual. The library was quantified using Qubit 1X dsDNAHS Assay Kits (Invitrogen, Q33230). Finally, the library was sequenced in a paired-end mode (PE150) on an Illumina NovaSeq 6000 platform (LC Bio Technology CO., Ltd., Hangzhou, China), using the NovaSeq 6000XP4-Lane Kit v1.5 (300 cycles) (Illumina, 20043131).

#### 2.2.2. Bioinformatics Analysis

The raw sequencing data in FASTQ format were quality controlled using fastp (https://github.com/OpenGene/fastp, version 0.23.4) with the parameters -1100-g-W6-5-q20-u 30. The sequencing data were then aligned to the host genome using Bowtie2 (https://www.sbgrid.org/software/titles/bowtie-2, version 2.2.0). Sequences aligned to the host genome were filtered out (this step is only necessary for projects involving host sequence filtering, such as human or mouse fecal samples), ensuring that subsequent assembly and analysis results were based solely on microbial sequences [11]. The effective sequences obtained were assembled using the software MEGAHIT (https://github.com/voutcn/megahit, version 1.2.9), producing FASTA format files where each sequence represented a contig. Contigs longer than 500 bp were retained, and coding region (CDS) prediction was conducted using MetaGeneMark (https://genemark.bme.gatech.edu/meta_gmhmmp.cgi, version 3.26), filtering out sequences with CDS lengths shorter than 100 nt. Following this, redundant sequences were removed using the software Mmseqs2 (https://github.com/soedinglab/Mmseqs2, version 15-6f452), generating non-redundant Unigenes. Clustering was performed at 95% identity and 90% coverage, with the longest sequence selected as the representative sequence to form the Unigenes set.

The effective sequences of each sample were aligned to the Unigene sequences using Bowtie2 (https://www.sbgrid.org/software/titles/bowtie-2, version 2.2.0), and the read counts for each Unigene were calculated. Unigenes with read counts less than or equal to 2 in all samples were filtered out. The final Unigenes set was used for subsequent analyses, with abundance statistics for each Unigene calculated. DIAMOND software (https://github.com/bbuchfink/diamond, version 0.9.14) was used to align the Unigene protein sequences with the NR_meta database to obtain species annotation information at various taxonomic levels, as well as annotations from functional databases such as KEGG (http://www.genome.jp/kegg, version 87.1), GO (http://geneontology.org, version 2018.12.21), eggNOG (http://eggnog5.embl.de/#/app/home, version 5.0), PHI (http://www.phi-base.org, version 4.14), CAZy (http://www.cazy.org, version 2022.08.06), CARD (https://card.mcmaster.ca/download, version v3.2.5), VFDB (http://www.mgc.ac.cn/VFs, version 2023.03.03), MGEs (https://github.com/KatariinaParnanen/MobileGeneticElementDatabase, version 2017.12.28), and BacMet (http://bacmet.biomedicine.gu.se, version 2.0), among others.

Finally, based on the abundance of Unigenes, the abundance of species and functional categories was calculated. Differential statistical analyses were performed between comparison groups based on species and function. For non-biological replicate samples, Fisher’s exact test was used, while for samples with biological replicates, the Wilcoxon rank-sum test (also known as the Mann–Whitney U test) was used for comparisons between two groups. For multiple groups with biological replicates, the Kruskal–Wallis test was applied. The significance threshold was set at *p* < 0.05 and |log2(fold_change)| > 1. Differential species or functional categories between groups were visualized using box plots created with R software (version 3.6.0).

Alpha diversity indices such as Chao1, Observed species, Goods coverage, Shannon, and Simpson were calculated for species at the genus level using QIIME1 (http://qiime.org/), and the results were visualized using rarefaction curves (R software, version 3.6.0). For samples with biological replicates greater than or equal to 5, inter-group differences were analyzed, and the results were displayed using violin and box plots. Statistical methods applied for two-group comparisons included the Wilcoxon rank-sum test (or Mann–Whitney U test), while for multi-group comparisons, the Kruskal–Wallis test was used. Beta diversity was calculated based on Bray–Curtis distance, and the results were visualized using PcoA (Principal Coordinates Analysis), NMDS (Nonmetric Multidimensional Scaling), UPGMA, Anosim (Analysis of Similarities), and Adonis (Permutational Multivariate Analysis of Variance) (R software, version 3.6.0).

LefSe (Linear Discriminant Analysis Effect Size) was used to identify differential species at various taxonomic levels between groups, with a threshold of LDA > 3.0 and *p* < 0.05. The results were visualized through evolutionary branching and bar charts. Spearman correlation analysis at the genus level was conducted using the R package ggplot2 (version 3.2.0) and ggnetwork, and relationships with |ho| > 0.8 were shown in network plots. Differential species analysis between two groups at the species level was performed using the R package metagenomeSeq (version 1.38.0).

### 2.3. Metabolomic Analysis

#### 2.3.1. Metabolite Extraction

Approximately 50 mg (±5 mg) of each sample was weighed and mixed with 500 μL of 80% pre-cooled methanol. The mixture was homogenized using a grinder, followed by incubation at −20 °C for 30 min. Samples were then centrifuged at 20,000× *g* for 10 min at 4 °C, and the supernatant was collected and centrifuged again for 5 min. The resulting supernatant was used for metabolite detection. In parallel, equal volumes from all samples were pooled to create a quality control (QC) sample.

Chromatographic Conditions: metabolite separation was performed on an ACQUITY UPLC HSS T3 column (100 mm × 2.1 mm, 1.8 μm, Waters, Milford, MA, USA). Mobile phase A consisted of 5 mmol/L ammonium acetate with 5 mmol/L acetic acid in water, and mobile phase B was acetonitrile. The gradient elution program was as follows:•0–0.8 min, 2–70% B;•0.8–2.8 min, 70–90% B;•2.8–5.3 min, 90–99% B;•5.3–5.9 min, 99% B;•5.9–7.5 min, 99–2% B;•7.5–7.6 min, 2% B;•7.6–10.0 min, 2% B.

The flow rate was 0.35 mL/min, the injection volume was 4 μL, and the column temperature was maintained at 40 °C.

Mass Spectrometry Conditions: an electrospray ionization (ESI) source was used in both positive and negative ionization modes, with data collected separately for each mode. The ion source temperature was set to 350 °C. The capillary voltage was +3.8 kV in positive mode and −3.4 kV in negative mode. The sheath gas flow rate was 50 Arb, auxiliary gas flow rate 15 Arb, and sweep gas flow rate 0 Arb. Data were acquired using full scan and data-dependent acquisition (DDA) modes on a Q-Exactive HF mass spectrometer (Thermo Fisher Scientific, Waltham, MA, USA). The full scan range was 70–1050 Da, with a resolution of 70,000, an automatic gain control (AGC) target of 3 × 10^6^, and a maximum ion injection time (Maximum IT) of 100 ms. The top five most intense ions (intensity > 100,000) from each full scan were selected for MS/MS (secondary) fragmentation, with a resolution of 17,500 and a Maximum IT of 50 ms. The dynamic exclusion duration was set to 6 s.

#### 2.3.2. Data Processing and Information Analysis

Raw mass spectrometry data were preprocessed using XCMS-V4.7 software, including peak detection, peak alignment, retention time correction, secondary peak grouping, and isotopic/adduct annotation. The LC-MS raw files were converted to mzXML format and processed using R software with the XCMS and metaX toolboxes. Ion features were identified based on retention time (RT) and mass-to-charge ratio (*m*/*z*). The intensity of each detected peak was recorded, generating a three-dimensional matrix comprising peak indices (RT–*m*/*z* pairs), sample names (observations), and ion intensities (variables).

Metabolite annotation was performed using the online KEGG and HMDB databases by matching the exact molecular mass (*m*/*z*) of detected features to reference data. Metabolites were annotated when the mass error between observed and database values was less than 10 ppm. Isotopic distribution patterns were further used to validate the molecular formulas. Additionally, an in-house database was used to verify metabolite identifications.

Statistical analysis of metabolomic data was performed primarily using R (version 4.0). Data preprocessing included filtering (removal of metabolites with >80% missing values across samples or >50% missing in QC samples), imputation (default method: KNN), and normalization (default method: PQN). Hierarchical clustering heatmaps were generated using the pheatmap package, while PCA and differential metabolite analyses were conducted with the metaX-v2.71 package. PLS-DA (Partial Least Squares Discriminant Analysis) was performed using the ropls package, and variable importance in projection (VIP) scores were calculated. Pearson correlation analysis was conducted using the cor function in R. Metabolites meeting the criteria of *p* < 0.05 (Student’s t-test), fold change (FC) > 1.2, and VIP ≥ 1 were defined as significantly differential metabolites.

Pathway enrichment of differential metabolites was performed using a hypergeometric test in the KEGG database. Pathways with a *p*-value < 0.05 were considered significantly enriched. In addition, Gene Set Enrichment Analysis (GSEA) (v4.1.0) was conducted to identify enriched KEGG pathways. Pathways with |NES| > 1, NOM *p*-value < 0.05, and FDR < 0.25 were considered significantly different between groups. Metabolic network diagrams were constructed based on the pathways in which the metabolites were Involved.

### 2.4. Data Analysis and Statistics

#### 2.4.1. Data Analysis

All statistical analyses were performed using R (version 4.0.0) and Python (version 3.7). Microbial community composition analysis, KEGG pathway enrichment, and CAZyme (Carbohydrate-Active Enzyme) activity analyses were visualized using the ggplot2 and pheatmap packages. Diversity analyses were conducted using QIIME2 and R to compare microbial community differences among different fermentation stages.

#### 2.4.2. Statistical Methods

The significance of differences among groups was assessed using ANOVA, with *p* < 0.05 considered statistically significant. When necessary, the Benjamini–Hochberg method was applied for multiple testing correction to reduce false positive results.

## 3. Results and Discussion

### 3.1. Sequencing Data, Microbial Richness, and Diversity

#### 3.1.1. α-Diversity Analysis

For a total of 18 compost substrate samples collected from six time points during the third fermentation process of *Agaricus bisporus*, α-diversity indices—including the Chao1 index, Observed Species index, Shannon index, Simpson index, and Goods_coverage index—were calculated (Table 1) to evaluate microbial community diversity across different fermentation stages.

To simulate resampling and assess trends in species variation and environmental richness, rarefaction curves were plotted. These curves quantify species richness and provide a visual comparison of microbial diversity among samples at different stages. The Goods_coverage index was used to evaluate microbial coverage, where higher values indicate a lower probability of undetected species in the sample. In this study, all samples achieved a Goods_coverage value of 1.0, confirming that the sequencing depth was sufficient and representative of the true microbial composition.

The Chao1 and Simpson Indices demonstrated that community richness and evenness remained at a high level throughout all fermentation stages, with the C stage exhibiting the highest species richness and evenness. Similarly, the Shannon index indicated a high degree of biodiversity and evenness across all stages, reflecting a stable and complex microbial ecosystem within the substrate [11].

#### 3.1.2. β-Diversity Analysis

Based on the taxonomic abundance profiles at different classification levels, Principal Component Analysis (PCA) was conducted to evaluate the variation in microbial community structure. The closer the samples appear in the PCA plot, the more similar their microbial compositions are. The results showed that microbial communities differed significantly among fermentation stages (Figure 1A).

To further validate these differences, we performed Analysis of Similarities (Anosim) to assess whether inter-group variation exceeded intra-group variation. The results revealed an R value of 0.4963 and a *p* value of 0.001, indicating that inter-group differences were significantly greater than intra-group differences and that the grouping was statistically meaningful and highly reliable (Figure 1B).

Overall, the compost substrate during the third fermentation cycle exhibited high microbial richness and evenness across all stages. Meanwhile, the microbial community composition displayed clear stage-specific differentiation, with the degree of divergence showing a positive correlation with the temporal progression of the fermentation process.

**Figure 1 microorganisms-13-02755-f001:**
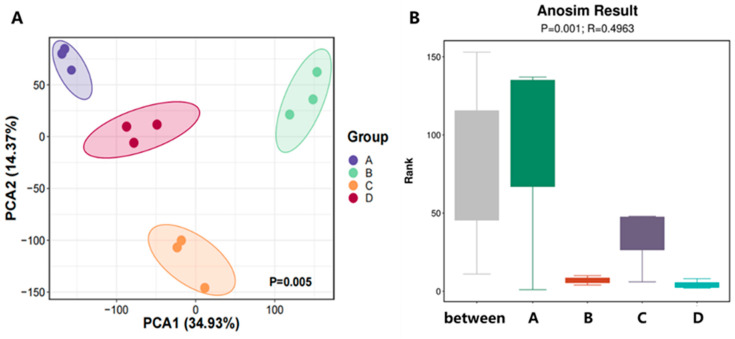
Microbial β-diversity of *Agaricus bisporus* substrates during the three successive fermentation processes. (**A**) Principal Component Analysis (PCA); (**B**) ANOSIM data analysis.

### 3.2. Microbial Community Succession and Composition During Agaricus bisporus Fermentation

Based on the Bray–Curtis distance matrix, Principal Coordinates Analysis (PcoA) was performed to evaluate microbial community changes across the four fermentation stages of *Agaricus bisporus*. The microbial communities of each stage were clearly separated, and the three replicates for each treatment clustered closely together, indicating good experimental reproducibility. Stages A, B, C, and D were significantly separated along the first principal coordinate (Pco1, 34.93%), while stages A and B exhibited a certain degree of similarity along the second principal coordinate (Pco2, 14.37%) (Figure 2).Figure 2Dynamic succession of microbial communities at different fermentation stages of *Agaricus bisporus*. (**A**) Phylum level. (**B**) Genus level. (**C**) Stage-specific microbes of *Agaricus bisporus* identified using LefSe across the four stages.
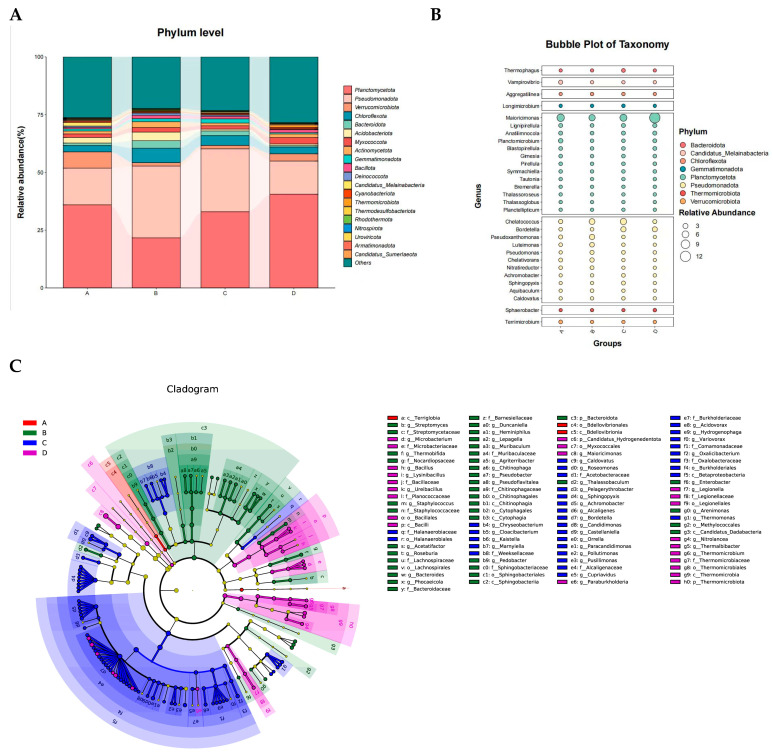


Microbial sequences were classified into 216 phyla, 249 classes, 519 orders, 1207 families, 4873 genera, and 36,715 species. Across all substrate samples, the dominant phyla during the four fermentation stages of *A. bisporus* were *Planctomycetota* (21.67–40.58%), *Pseudomonadota* (14.35–31.07%), *Verrucomicrobiota* (1.47–7.06%), *Chloroflexota* (2.75–6.29%), and *Bacteroidota* (0.59–3.27%), collectively accounting for over 60% of all sequences. The top ten phyla additionally included *Acidobacteriota* (0.73–3.71%), *Myxococcota* (1.43–2.78%), *Actinomycetota* (1.18–2.55%), *Gemmatimonadota* (0.58–1.78%), and *Bacillota* (0.65–1.38%), all of which were detected in all substrate samples with relative abundances below 5% in each layer (Figure 2A).

During the mid-fermentation stage (B stage, day 10), the relative abundances of *Pseudomonadota*, *Chloroflexota*, *Bacteroidota*, *Acidobacteriota*, *Actinomycetota*, and *Bacillota* increased significantly, whereas those of *Planctomycetota* and *Verrucomicrobiota* decreased. *Bacillota*, commonly present in composting substrates, can survive high temperatures, and its higher abundance reflects accelerated cellulose degradation during the B stage [12,13,14,15]. *Actinomycetota* play a critical role in organic matter biodegradation, with their abundance positively correlating with cellulase activity [16,17]. In the late fermentation stages (C and D), *Planctomycetota* gradually dominated, likely due to their carbohydrate-active enzyme (CAZymes) capabilities, which are involved in the breakdown of complex *polysaccharides* [18]. Meanwhile, *Pseudomonadota* and *Bacteroidota* play key roles in organic matter decomposition and carbon cycling, with *Bacteroidota* able to convert lignocellulose into short-chain fatty acids [19,20].

At the genus level (Figure 2B), the ten dominant genera across all substrate samples were *Maioricimonas* (5.07–12.14%), *Chelatococcus* (1.35–5.09%), *Bordetella* (0.62–3.43%), *Pseudoxanthomonas* (0.05–3.41%), *Lignipirellula* (0.82–1.57%), *Anatilimnocola* (0.77–1.39%), *Planctomicrobium* (0.58–2.10%), *Luteimonas* (0.41–2.37%), *Blastopirellula* (0.54–1.03%), and *Pseudomonas* (0.32–1.53%), collectively accounting for approximately 19% of all species. Among these, *Maioricimonas* decreased in relative abundance during stage B but increased in stages C and D, eventually becoming dominant. *Maioricimonas*, originally isolated from the marine sponge *Pseudoceratina* sp., is a pink, *halotolerant*, and *alkalitolerant Planctomycetota* strain with biotechnological potential in industrially relevant enzymes such as sulfatases and CAZymes [21]. Conversely, genera within *Pseudomonadota*, including *Chelatococcus*, *Pseudoxanthomonas*, and *Luteimonas*, exhibited an initial increase followed by a decrease as fermentation progressed. Notably, *Bordetella* maintained its relative abundance during stages C and D and is primarily associated with the degradation of lignin-derived compounds in environmental contexts [22].

Using Linear Discriminant Analysis Effect Size (LefSe) with an LDA score threshold of 2 (Figure 2C), stage-specific microbial taxa were identified across the four fermentation stages. The core microbial communities differed significantly between stages. In stage A, only a single class and order, *Bdellovibrionia* and *Bdellovibrionales*, were identified as biomarkers. Stage B exhibited 1 phylum, 4 classes, 5 orders, 8 families, and 17 genera, predominantly including *Chitinophagaceae* and *Muribaculaceae*. Stage C, the late fermentation stage, displayed the greatest number of indicator taxa, comprising 1 class, 1 order, 7 families, and 24 genera, with core genera including *Bordetella*, *Achromobacter*, and *Hydrogenophaga*. Stage D included 2 phyla, 2 classes, 4 orders, 5 families, and 8 genera, with core genera mainly represented by *Ureibacillus* and *Microbacterium*.

In summary, PcoA revealed a pronounced succession of microbial community structures during the four fermentation stages (A–D) of A. bisporus (Pco1 explained 34.93%). Dominant phyla included *Planctomycetota*, *Pseudomonadota*, *Verrucomicrobiota*, *Chloroflexota*, and *Bacteroidota*, collectively accounting for over 60% of all sequences. Fermentation drove dynamic changes in key functional taxa: during mid-fermentation (stage B, day 5), *Pseudomonadota*, *Bacillota*, and *Actinomycetota* were significantly enriched, with increased abundances strongly associated with enhanced cellulose degradation (*p* < 0.05). In the late fermentation stages (C/D, days 10/15), *Planctomycetota*, particularly *Maioricimonas*, became dominant, with relative abundance increasing by 138%, primarily participating in complex polysaccharide transformation. LefSe analysis further identified stage-specific biomarker taxa (e.g., *Chitinophagaceae* and *Muribaculaceae* at stage B; *Bordetella*, *Achromobacter*, and *Hydrogenophaga* at stage C; *Ureibacillus* and *Microbacterium* at stage D), which exhibited strong functional synergy with lignocellulose degradation modules.

Collectively, the fermentation stage is a critical driver of microbial community restructuring and functional differentiation, with stage-specific taxa playing pivotal roles in cellulose degradation and polysaccharide transformation within the *A. bisporus* substrate.

### 3.3. Stage-Specific Metabolic Shifts in CARBON Source Utilization During Agaricus bisporus Fermentation

The study of the abundance of CAZymes (carbohydrate-active enzymes) involved in microbial community degradation and the correlation analysis of carbohydrate gene temporal expression, especially the changes in enzyme content and correlation during cellulose and lignin degradation, reveals the mechanisms of cellulose and other complex carbohydrate degradation.

A total of 586 different CAZyme families were detected among the annotated genes, including 293 GHs (glycoside hydrolases), 103 GTs (glycosyltransferases), 74 PLs (polysaccharide lyases), 72 CBMs (carbohydrate-binding modules), 24 Aas (auxiliary activities), and 20 Ces (carbohydrate esterases). During *Agaricus bisporus* fermentation, the most abundant enzymes were GHs and GTs. At the family level, GHs, GTs, and Aas were particularly abundant throughout the fermentation process. Specifically, during stages B and C, GT35 (glycosyltransferases), GT2 (cellulose/chitin synthases) [23], GT4 (sucrose synthases) [24], and GT1 (nucleotide transferases) [25] were the most abundant, while in stages B and D, GH1 (cellulose/lactase) [26,27] and GH23 (peptidoglycan hydrolases, lysozymes) [28] showed the highest concentrations (Figure 3).

Furthermore, Aas, which assist in the oxidation-reduction processes for carbohydrate degradation (especially for complex polysaccharides like cellulose and lignin), were highly abundant during stages B, C, and D. This suggests that the fermentation process of *Agaricus bisporus* is predominantly driven by enzymes involved in cellulose degradation, with enzymes related to lignin degradation playing a supplementary role in the breakdown of complex carbohydrates.

The temporal changes In carbohydrate-related genes during fermentation demonstrated the differences in gene expression between stages AB and CD, revealing microbial community shifts and corresponding metabolic transitions, particularly in the degradation of cellulose and other complex carbohydrates. During fermentation, the content of CBM families increased significantly. CBMs are non-catalytic domains that bind specific substrates to assist enzymes in their functions, without catalyzing reactions. In stage AB, the most abundant CBMs were CBM2 (binding substrates such as cellulose, β-1,3-glucans) [29], CBM38 (binding substrates such as chitin, chitosan), CBM51 (binding substrates such as arabinoxylan, xylan) [30], and CBM57 (binding substrates such as crystalline cellulose, xylan) [31]. In this stage, enzymes from the GH1 family, associated with cellulose or hemicellulose degradation, were also abundant, and a strong positive correlation was observed between the two, indicating a strong enzymatic interaction. Through CBM57 and CBM2, enzymes bind to cellulose and xylan, localizing to the cellulose-xylan complex’s surface, indirectly promoting cellulose and hemicellulose degradation. Additionally, CBM6 can bind hemicellulose (such as xylan), which is often covalently linked to lignin, thus helping enzymes localize near lignin.

In the CD stage, except for the CBM57 content associated with cellulose degradation, which remained at similar levels as in stage AB, the content of CBM6 significantly increased (Figure 4). This indicates that, throughout the fermentation process, there was extensive cellulose degradation, accompanied by lignin degradation, resulting in the same outcome as the heatmap conclusion, where cellulose degradation predominated, and lignin degradation was a supplementary process.

Cellulose, hemicellulose, and lignin are the main components of lignocellulosic biomass [32]. In this study, cellulose and hemicellulose-degrading enzymes exhibited the highest activity during *Agaricus bisporus* fermentation. The GH1 family was the primary cellulose-degrading enzyme during stage AB, whereas in stages CD, enzymes from the CBM57 family and others that bind substrates participated in cellulose degradation. Simultaneously, lignin degradation occurred as well. Although white-rot fungi in Basidiomycetes are known to degrade lignin more rapidly and extensively than other microorganisms [33], typically attributing lignin degradation to their metabolic processes, this study demonstrates that *Agaricus bisporus*, as a saprotrophic fungus, plays a key role in lignin degradation.

### 3.4. Key Functional Microbial Communities Drive the Construction of Metabolic Networks

A microbial network based on significant correlations (Spearman correlation) was established for the two fermentation stages of *Agaricus bisporus* (AB and CD stages). In the AB stage, the network consisted of 50 nodes and 1182 links, while in the CD stage, the microbial network contained 50 nodes and 608 links. In the correlation network analysis of the AB stage (Figure 5A), the top five dominant genera were *Acidobacteriota_bacterium*, *Lignipirellula_cremea*, *Verrucomicrobiaceae_bacterium*, *Aggregatilinea*_sp., and *Planctomycetaceae_bacterium*, with 34 (14 positive and 20 negative), 34 (21 positive and 13 negative), 34 (21 positive and 13 negative), 34 (13 positive and 21 negative), and 33 (22 positive and 12 negative) links, respectively. In the CD stage, the relationships between species were further clarified, especially the strong correlations among *Maioricimonas rarisocia*, *Anatilimnocola_aggregata*, *Planctomycetia_bacterium_21.64.5*, *Panctomycetes_bacterium_RBG_16_64_12*, *Blastopirellula_marina*, and *Planctomycetes_bacterium_RBG_13_63_9*, indicating their synergistic interaction in the later fermentation stages, further promoting the degradation of cellulose and lignin. Notably, *Bordetella* was only involved in a correlation network with *Anatilimnocola* and itself (Figure 5B).

In the AB stage (Figure 5A), the network was more complex, comprising 50 nodes and 1182 links, indicating intense and non-specific interactions among a diverse microbial community as they initiated substrate colonization and primary degradation. In stark contrast, the CD stage network (Figure 5B) was significantly streamlined (50 nodes and 608 links), suggesting a shift towards a more specialized and efficient synergistic community, likely focused on the degradation of recalcitrant lignocellulosic components.

During *Agaricus bisporus* fermentation, the microbial community’s ability to degrade cellulose directly impacts the efficiency of fermentation. The microbial communities related to cellulose degradation exhibited clear succession across the different fermentation stages (A, B, C, and D). According to the analysis of species related to K01179 (endo-β-glucanase) (Figure 5C) and K01186 (β-glucosidase) (Figure 5D), species abundance showed a trend of decreasing and then increasing across the fermentation stages. Based on K01186, the microbial community was more diverse in stage A, and cellulose-degrading species did not dominate. However, as fermentation progressed, particularly in stage B, the abundance of most species decreased, leading to a significant increase in the relative abundance of *Maioricimonas rarisocia*, *Lentisphaerae* bacterium RIFOXYC12, and *Fulvivirga ligni*, demonstrating their key roles in cellulose degradation. In the later stages of fermentation, *Maioricimonas rarisocia* became dominant, indicating a close correlation between its abundance and cellulose degradation capacity, particularly as the microbial community adapted to the degradation of complex carbon sources, with cellulose degradation becoming a core function throughout the fermentation process [34].

At the same time, lignin is one of the most complex carbon sources in plants, and the microbial community’s ability to degrade lignin also significantly influences fermentation efficiency. According to the species analysis related to K01183 (lignin peroxidase) (Figure 5E) and K01193 (aromatic compound degradation pathway) (Figure 5F), the overall microbial abundance was lower than that for K01179 and K01186, indicating that lignin degradation may be a secondary function throughout the fermentation process. In different fermentation stages, species abundance also showed a trend of decreasing and then increasing. Based on K01193, *Maioricimonas rarisocia* dominated throughout the fermentation process.

The third fermentation process of *Agaricus bisporus* exhibited a clear functional reshaping pattern of the microbial community. As shown in Figure 5A, the complex competitive network (1182 links) of the AB stage evolved into a streamlined cooperative network (608 links) in the CD stage, marking the specialization of the *Planctomycetes* functional community. Gene correlation analysis (Figure 5D) further confirmed that cellulose degradation was predominantly driven by *M. rarisocia*, with its abundance surge strongly positively correlated with β-glucosidase activity (r = 0.89). Lignin degradation, as a subordinate pathway, was fully dependent on the metabolic capacity of *M. rarisocia*. This phenomenon highlights the dual-functional engine role of *M. rarisocia*—enhancing carbon source conversion efficiency through the degradation of both cellulose (K01186) and lignin-derived aromatic compounds (K01193), thus promoting metabolic network optimization in the later stages of fermentation. The dynamic abundance of this strain was significantly negatively correlated with network simplification (AB→CD reduction by 48.6%) (r = −0.76), suggesting that functional specialization is the core mechanism for efficient degradation.

### 3.5. Implications of Microbial Succession for Agaricus bisporus Growth

The stage-specific microbial succession we observed raises the question of its functional relationship with the growth of *A. bisporus*. We hypothesize a two-phase interaction model. In the pre-fruiting fermentation stages (A–C), the microbial consortium, particularly the cellulolytic pioneers in stage B and the complex polysaccharide degraders like *Maioricimonas* in stage C, acts as a “bio-preprocessor.” They break down lignocellulose into more accessible oligo- and mono-saccharides, which likely facilitates subsequent nutrient assimilation by *A. bisporus* after spawning.

Furthermore, the well-established microbial network in stage D (Figure 5B) may create a protective “coconut” that competitively excludes pathogens, thus securing a favorable niche for *A. bisporus* colonization. While our metagenomic data primarily reflect the metabolic potential of the prokaryotic community, the observed successional pattern—from a diverse and competitive network to a simplified and synergistic one—suggests a community assembly process that ultimately paves the way for the mushroom fungus to thrive. Future research co-cultivating the key bacterial isolates identified here (e.g., *Maioricimonas*) with *A. bisporus* will be crucial to validate these potential synergies or competitions for space and nutrients.

## 4. Conclusions

This study systematically reveals the synergistic mechanisms of microbial community succession and carbon source metabolic functions during the third fermentation process of *Agaricus bisporus* (AB). By integrating metagenomic and metabolomic analyses, the study clarifies the microbial community dynamics across four fermentation stages (A: inoculation; B: fermentation completion, fruiting bed formation; C: B + 5 days; D: C + 5 days), as shown by the Bray–Curtis distance-based PCoA analysis, which clearly demonstrates significant separation of the microbial communities (the first principal component explained 34.93% of the variance). During stage B, the enriched cellulose-degrading phyla *Pseudomonadota* (peak relative abundance 31.07%) and *Actinomycetota* (peak 2.55%) were positively correlated with cellulase activity, directly driving the rapid degradation of cellulose. In the later stages (C/D), *Planctomycetota* (abundance 21.67–40.58%) dominated the degradation of complex polysaccharides through its CAZyme metabolic capacity. The genus *Maioricimonas* showed an increase in abundance to 12.14% in stages C/D, and along with the LEfSe-identified marker species, such as *Bordetella* (degrading lignin derivatives) in stage C, together formed the lignin degradation functional module.

At the carbon metabolism level, stage B was dominated by cellulose-degrading enzymes, with peak abundances of GT2 (cellulose/chitin synthase), GT4 (sucrose synthase), and GH1 (cellulose/lactase), which directly accelerated carbon source conversion. During the AB stage, the CBM family (such as CBM2 and CBM57) assisted enzyme localization through binding to cellulose-hemicellulose complexes, indirectly enhancing degradation efficiency. In contrast, the significant increase in CBM6 abundance in the CD stage indicates a shift in metabolic focus toward lignin degradation. CBM6 localizes lignin structures through binding to hemicellulose (such as xylan), thus expanding the scope of degradation.

The microbial interaction network further validated the stage-specific functional transitions: In stage AB, core genera (such as *Lignipirellula*) formed an efficient cellulose degradation network with 1182 links, whereas in stage CD, the network simplified to 608 links, with the synergistic action of *Maioricimonas* and *Bordetella* dominating the K01193 (aromatic compound degradation) pathway. Their abundance was directly correlated with lignin degradation efficiency. The analysis of K01186 (β-glucosidase) and K01193 pathways confirmed that *Maioricimonas rarisocia* dominates carbon cycling throughout fermentation, underscoring its core role in lignocellulosic resource utilization.

These findings provide the first elucidation of the stage-specific metabolic transition mechanisms during *Agaricus bisporus* fermentation. Stage B relies on cellulose-degrading microbiota (e.g., *Pseudomonadota/Actinomycetota*) and GHs enzyme systems for efficient carbon conversion, while stages C/D are driven by *Planctomycetota* (e.g., *Maioricimonas*) through CAZymes, facilitating lignin degradation. Collectively, this microbial succession not only degrades agricultural waste efficiently but also likely primes the substrate and structures a beneficial microbiome for the subsequent cultivation of A. bisporus. This mechanism offers a theoretical foundation for optimizing fermentation processes (e.g., targeted inoculation strategies or engineered CAZyme systems) and opens new pathways for the biocircular utilization of agricultural waste. Future research could focus on the directed engineering of CAZymes to enhance lignocellulose degradation efficiency on an industrial scale.

## Figures and Tables

**Figure 3 microorganisms-13-02755-f003:**
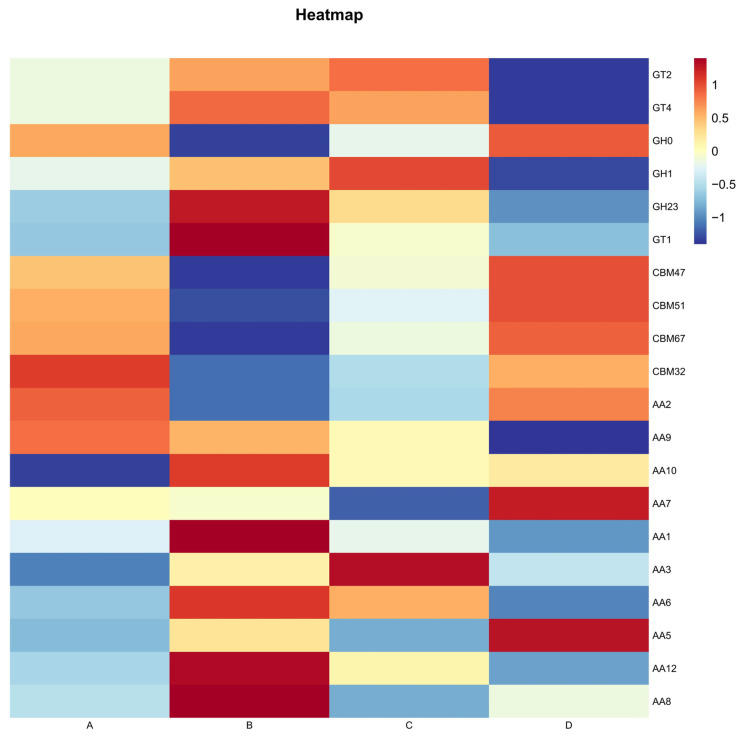
CAZy Level 2 heatmap for each group.

**Figure 4 microorganisms-13-02755-f004:**
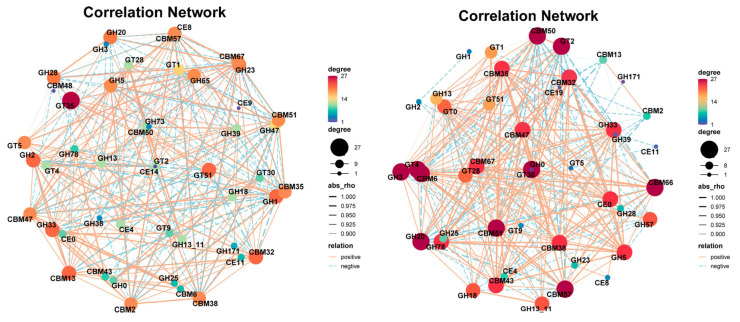
Temporal changes in carbohydrate-related genes during AB and CD stages.

**Figure 5 microorganisms-13-02755-f005:**
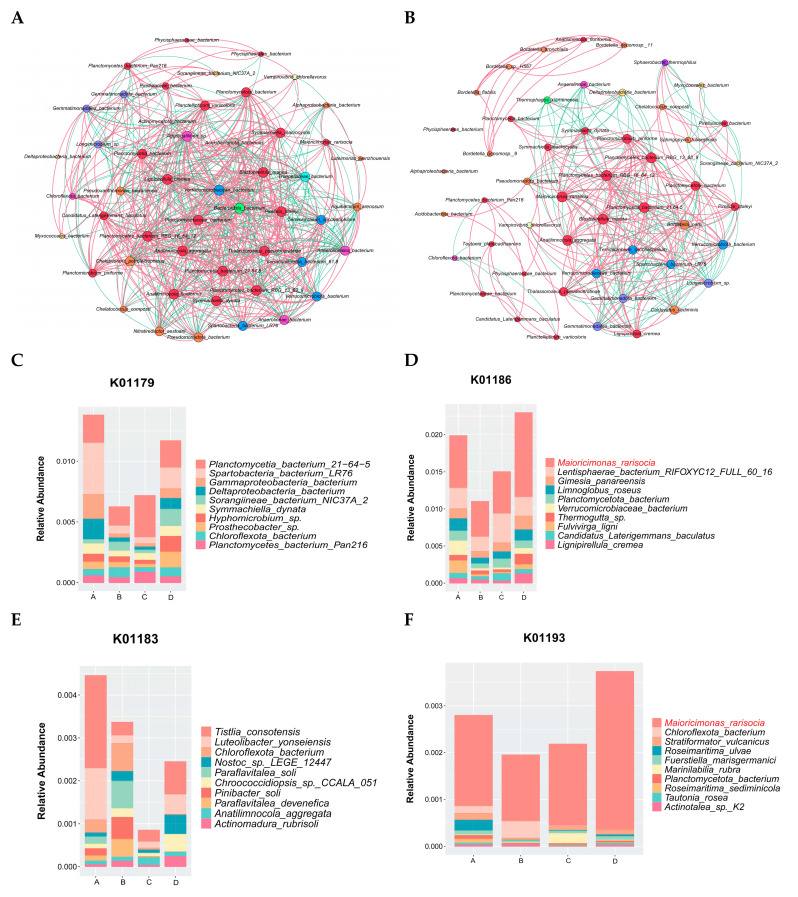
Microbial community network analysis at different stages of *Agaricus bisporus* fermentation. The node size is proportional to the genus abundance. The color of the nodes corresponds to the taxonomic classification of the genus. Edge colors represent positive correlations (red) and negative correlations (green). (**A**) AB stage. (**B**) CD stage. Microbial enrichment in cellulose and lignin pathways. (**C**) K01179. (**D**) K01186. (**E**) K01183. (**F**) K01193.

**Table 1 microorganisms-13-02755-t001:** Summary of α-diversity indices across all *Agaricus bisporus* fermentation samples.

	Observed_Species	Shannon	Simpson	Chao1	Goods_Coverage
A1	24,482.00	7.90	0.97	24,805.28	1.00
A2	22,225.00	7.60	0.97	22,817.72	1.00
A3	22,035.00	7.56	0.97	22,681.69	1.00
B1	25,023.00	8.00	0.97	25,295.00	1.00
B2	25,034.00	8.03	0.97	25,336.99	1.00
B3	24,952.00	8.07	0.97	25,221.23	1.00
C1	24,202.00	7.67	0.97	24,532.71	1.00
C2	24,075.00	7.99	0.97	24,485.50	1.00
C3	24,179.00	7.69	0.97	24,592.92	1.00
D1	23,696.00	7.60	0.97	24,109.48	1.00
D2	23,853.00	7.73	0.97	24,241.74	1.00
D3	23,370.00	7.56	0.96	23,846.73	1.00

## Data Availability

The original contributions presented in this study are included in the article. Further inquiries can be directed to the corresponding authors.

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
