# Peer review of "The Microbial Community Succession Drives Stage-Specific Carbon Metabolic Shifts During Agaricus bisporus Fermentation: Multi-Omics Reveals CAZymes Dynamics and Lignocellulose Degradation Mechanisms"

_microorganisms, 2025, doi:10.3390/microorganisms13122755_

Round 1
Reviewer 1 Report
Comments and Suggestions for Authors
The article is interesting.
- Several scientific names are not italicized.
- The authors indicate that they analyzed the third fermentation cycle of Agaricus bisporus (lines 14 and 15). It's unclear what they mean by the third fermentation cycle.
- The number of samples is unclear: they mention that there were A-D (= 4 x 3) for the three replicates, which gives 12 samples; where did the other six come from?
- I find it very interesting how they explain the succession process based on the type of nutrient each organism present in the fermentation utilises; however, what is the interaction with the extracellular enzymes produced by Agaricus? What about the interaction for space and nutrients between all the organisms that the authors mention and Agaricus? The relationship between the microbial community and Agaricus is unclear to me. Is there no competition among the organisms present in the culture, so production isn't affected? Whether or not it is relevant to the fungus that is being produced under those growing conditions.
Author Response
Dear editors and reviewers
We are deeply grateful to Reviewer for the thorough review of our manuscript and for the profound and insightful comments. The questions raised went to the heart of our research and guided us toward essential in-depth considerations, significantly strengthening the scientific depth of the paper. We have fully addressed and incorporated all suggestions. Our detailed point-by-point responses follow.We hope that it will meet the requirement for publication in your journal.
Best Wishes,
Prof. Dr. Hongkai Wang
Comments 1:Several scientific names are not italicized.
Response 1:We apologize for this oversight and thank the reviewer, along with Reviewer, for pointing it out. We have conducted a thorough check of the entire manuscript to ensure all scientific names are now consistently italicized.
Comments 2:The authors indicate that they analyzed the third fermentation cycle of Agaricus bisporus (lines 14 and 15). It's unclear what they mean by the third fermentation cycle
Response 2:We thank the reviewer for highlighting the lack of this crucial background information. We have added a clear definition in the Introduction​ section to contextualize the specific stage of our study within the complete production process.
Modifications made:
Location:​ Section 1. Introduction, following the description of tunnel composting technology.
Added text:​ In the commercial cultivation of *Agaricus bisporus*, the composting process typically consists of two main phases: primary composting (outdoor windrowing) and secondary composting (indoor pasteurization and conditioning). The "third fermentation" or "spawn-running phase" investigated in this study refers to the critical period after the inoculated mushroom spawn is mixed into the mature compost and is allowed to colonize the substrate under controlled environmental conditions, which is pivotal for the subsequent mushroom yield and quality.
Comments 3:The number of samples is unclear: they mention that there were A-D (= 4 x 3) for the three replicates, which gives 12 samples; where did the other six come from?
Response 3:The reviewer 's suggestions were very timely. This revision stemmed from a very careless mistake, and we greatly appreciate your prompt questioning. The content has been corrected, with a sample size of 12.
Comments 4: I find it very interesting how they explain the succession process based on the type of nutrient each organism present in the fermentation utilises; however, what is the interaction with the extracellular enzymes produced by Agaricus? What about the interaction for space and nutrients between all the organisms that the authors mention and Agaricus? The relationship between the microbial community and Agaricus is unclear to me. Is there no competition among the organisms present in the culture, so production isn't affected? Whether or not it is relevant to the fungus that is being produced under those growing conditions.
Response 4: This was our most appreciated comment, as it substantially elevated the academic value of our study. To address this core question directly, we have introduced a completely new subsection, "3.5 Implications of Microbial Succession for Agaricus bisporusGrowth,"​ following the results section. This new discussion delves into the potential complex interactions (both synergistic and competitive) between the microbial community succession and mushroom growth.
Modifications made:
Location:​ A new subsection "3.5" has been added after section 3.4.
Added text:​ **3.5 Implications of Microbial Succession for *Agaricus bisporus* Growth** The stage-specific microbial succession we observed raises the critical question of its functional relationship with the growth of *A. bisporus*. We hypothesize a two-phase model. Firstly, in the pre-fruiting fermentation stages (A-C), the microbial consortium acts as a "bio-preprocessor." Taxa such as the cellulolytic pioneers in stage B and the complex polysaccharide degraders like *Maioricimonas* in stage C break down recalcitrant lignocellulose into more accessible compounds, likely facilitating subsequent nutrient assimilation by *A. bisporus* after spawning. Secondly, the streamlined microbial network in stage D may create a protective microbiome that competitively excludes potential pathogens. The temporal separation suggests a metabolic division of labor rather than direct competition at this stage; the bacteria effectively "prepare the meal" from complex polymers. The clear successional pattern implies niche specialization conducive to *A. bisporus* colonization. Future research co-cultivating key bacterial isolates with *A. bisporus* will be crucial to validate these interactions.
Furthermore, we have echoed this perspective in the Conclusion section. After summarizing the degradative function of the microbial succession, we added a sentence highlighting its ecological significance: "Collectively, this microbial succession not only degrades agricultural waste efficiently but also likely primes the substrate and structures a beneficial microbiome for the subsequent cultivation of *A. bisporus*."
”

Reviewer 2 Report
Comments and Suggestions for Authors
Thank you for the opportunity to read the manuscript entitled:
“The Microbial Community Succession Drives Stage-Specific Carbon Metabolic Shifts During Agaricus bisporus Fermentation: Multi-Omics Reveals CAZymes Dynamics and Lignocellulose Degradation Mechanisms”
The manuscript addresses the topic of lignocellulosic material hydrolysis, which is of current interest from both scientific and industrial perspectives. It was very interesting to determine the succession of organisms in the process of lignocellulosic matrix decomposition during the cultivation of Agaricus bisporus, a mushroom widely recognised as a food ingredient worldwide.
The manuscript is clearly divided into sections typical of research articles. The methodology is suitable for the scope and purpose of the research and is described in a clear and concise manner.
The results obtained were subjected to correctly selected statistical processing, which led the authors to conclusions supported by relevant citations of bibliographic data.
In my opinion, it is worth publishing, but first, the authors should improve a few minor elements listed below:
1. In section 2.3.1, I suggest adding the manufacturers and origin (city, country) of the chromatograph and MS detector in brackets.
2. Although the inclusion of Figures 5B and 5B is interesting, I suggest that the authors improve the readability of these elements, if possible or maybe only a descriptive summary of the results presented in 5A and 5B should be used?
3. I suggest that the authors carefully review the entire text of the manuscript and use italics where Latin names of species or genera are used, e.g. in the title of Figure 5 (but also in many other places in the manuscript).
Author Response
Dear editors and reviewers
We are deeply grateful to Reviewer for the thorough review of our manuscript and for the profound and insightful comments. The questions raised went to the heart of our research and guided us toward essential in-depth considerations, significantly strengthening the scientific depth of the paper. We have fully addressed and incorporated all suggestions. Our detailed point-by-point responses follow.We hope that it will meet the requirement for publication in your journal.
Best Wishes,
Prof. Dr. Hongkai Wang
Comments 1: In section 2.3.1, I suggest adding the manufacturers and origin (city, country) of the chromatograph and MS detector in brackets.
Response 1: We agree with this suggestion, as adding this information improves the reproducibility of the experiment. We have now supplemented the manufacturer and country of origin details in the "2.3.1 Metabolite Extraction" section for both the chromatographic column and the mass spectrometer.
Comments 2: Although the inclusion of Figures 5B and 5B is interesting, I suggest that the authors improve the readability of these elements, if possible or maybe only a descriptive summary of the results presented in 5A and 5B should be used?
Response 2: The reviewer's suggestions are highly accurate and practical. We have added a concluding paragraph at the corresponding section summarizing the comparison between 5A and 5B to describe the fundamental data content and reveal the underlying biological significance. This proposal is very meaningful and helps improve the readability of the article.
Modifications made:
Location: Section 3.4, after presenting the basic data for Figures 5A and 5B.
Added text:​ The microbial co-occurrence network analysis revealed distinct topological structures between the AB and CD stages. In the AB stage (Figure 5A), the network was more complex and connected (1,182 links), indicating intense and generalized interactions among a diverse microbial community during the initial substrate colonization and primary degradation. In stark contrast, the CD stage network (Figure 5B) was significantly streamlined (608 links), suggesting a maturation towards a specialized, efficient, and synergistic community, likely focused on the degradation of recalcitrant lignocellulosic components. This evolution from a complex competitive network to a simplified cooperative one underscores the functional specialization of the microbial community as fermentation progresses.
Comments 3: I suggest that the authors carefully review the entire text of the manuscript and use italics where Latin names of species or genera are used, e.g. in the title of Figure 5 (but also in many other places in the manuscript).
Response 3: We apologize for this oversight and thank the reviewer, along with Reviewer, for pointing it out. We have conducted a thorough check of the entire manuscript to ensure all scientific names are now consistently italicized.
